# Histopathology and Phylogeny of the Dinoflagellate *Hematodinium perezi* and the Epibiotic Peritrich Ciliate *Epistylis* sp. Infecting the Blue Crab *Callinectes sapidus* in the Eastern Mediterranean

**DOI:** 10.3390/microorganisms12030456

**Published:** 2024-02-23

**Authors:** Athanasios Lattos, Dimitrios K. Papadopoulos, Ioannis A. Giantsis, Alexios Stamelos, Dimitrios Karagiannis

**Affiliations:** 1Laboratory of Ichthyology–Culture and Pathology of Aquatic Animals, Faculty of Veterinary Medicine, Aristotle University of Thessaloniki, 541 24 Thessaloniki, Greece; astamel@vet.auth.gr (A.S.); dkaragiannis@vet.auth.gr (D.K.); 2Department of Animal Science, Faculty of Agricultural Sciences, University of Western Macedonia, 531 00 Florina, Greece; dkpapado@bio.auth.gr

**Keywords:** *Callinectes sapidus*, *Hematodinium perezi*, mass mortality, bioinvasion, infectious diseases, dinoflagellates, parasites

## Abstract

Bioinvasions constitute both a direct and an indirect threat to ecosystems. Direct threats include pressures on local trophic chains, while indirect threats might take the form of an invasion of a microorganism alongside its host. The marine dinoflagellate *Hematodinium perezi*, parasitizing blue crabs (*Callinectes sapidus*), has a worldwide distribution alongside its host. In Greece, fluctuations in the blue crab population are attributed to overexploitation and the effects of climate change. The hypothesis of the present study was that blue crab population reductions cannot only be due to these factors, and that particular pathogens may also be responsible for the fluctuations. To investigate this hypothesis, both lethargic and healthy blue crab specimens were collected from three different fishing sites in order to assess the health status of this important species. Together with the lethargic responses, the hemolymph of the infested crabs presented a milky hue, indicating the first signs of parasitic infestation with *H. perezi*. The histopathological results and molecular identification demonstrated the effect of the presence of *H. perezi* in the internal organs and their important role in the mortality of blue crabs. Specifically, *H. perezi*, in three different tissues examined (heart, gills, hepatopancreas), affected the hemocytes of the species, resulting in alterations in tissue structure. Apart from this dinoflagellate parasite, the epibiotic peritrich ciliate *Epistylis* sp. was also identified, infecting the gills. This study represents the first detection of *H. perezi* in the eastern Mediterranean, demonstrating that this is the main causative agent of blue crab mortality on Greek coastlines.

## 1. Introduction

Global invasions, i.e., the successful establishment, breeding, and spreading of organisms outside of their natural range, may have detrimental consequences on local ecosystems. Invasions in aquatic ecosystems are increasingly frequent and are responsible for pressure on local habitats, posing a risk to biodiversity by spreading pathogenic microorganisms to the new ecosystems [1,2]. Climate change, shipping by the means of ballast waters, global trade, and accidental or intentional (for rearing purposes) releases of non-indigenous species into new ecosystems have intensified the phenomenon in recent years, affecting local flora and fauna [3,4,5,6,7,8]. In parallel with aquatic animals, aquatic pathogens may also invade new ecosystems together with their host, representing another important threat to ecosystems [4,7]. Crustaceans constitute one of the most important examples of global invasions, with more than 240 species recorded and characterized as non-indigenous in the Mediterranean Sea [9].

The Atlantic blue crab, *Callinectes sapidus* Rathbun, 1986 (Brachyura: Portunidae), is a highly mobile benthic predator distributed from Nova Scotia, Canada, to Brazil, economically supporting fisheries in the local coastal areas [10,11,12,13]. Beyond its large distribution range of the species in Atlantic coastlines, it has been reported as invader in the Mediterranean Sea [11,12]. *C. sapidus* invasion was first reported in France in 1900, likely resulting from either ballast waters or having been imported for aquaculture purposes [11,14,15]. Since then, it has invaded many habitats around the Mediterranean Sea, as well as in the Black Sea [11,12]. *C. sapidus* prefers enclosed environments such as estuaries or lagoons between 0 m and 90 m in depth, mostly on sandy or muddy substrates, regardless of the presence of vegetation [16]. Despite its important status in fisheries in the USA, its exploitation in the Mediterranean Sea is limited mostly to Turkey, Greece, Italy, and Egypt [17,18,19,20]. The blue crab diet consists of a broad spectrum of prey, including mollusks, crustaceans, fish, polychaetes, and marine vegetation [21]. Although the blue crab constitutes an important model organism for studying invasions related to the spread of microorganisms, the majority of scientific research is focused on the population dynamics, structure, and fisheries of the species [12].

During recent years, the blue crab populations across Greek coastlines have been observed to decline, probably on account of high human exploitation and overfishing, resulting in massive population reductions in several circumstances. Additionally, urban pollution and agricultural waste that ends up in marine habitats exerts pressure on the populations and affects the income of many professional fishermen [22]. Regardless of the direct human pressures on habitats and on the organism, mass mortality of the species has emerged, causing a significant deterioration of the populations in Greek coastlines. Mortalities occur mostly in spring and autumn, causing tons of dead individuals to wash ashore in the lagoons or gulfs of the Aegean and Ionian Seas. These phenomena, resulting in blue crab population fluctuations, have only been attributed to the aforementioned factors, i.e., overfishing and pollution. The presence of pathogens, however, has not been investigated so far, even though crustaceans are particularly vulnerable to various parasites [10]. It should also be noted that invasive marine animals may be symptomatic or asymptomatic carriers of invasive microorganisms as well, worsening the negative effects of invasions to biodiversity.

Thus, the main objective of the current research was to investigate blue crab population reductions and mortalities as well as to identify any potential pathogen as the causative agent of the declines in this valuable fisheries resource. Although mortalities of this species occur in several locations in Greece each year, no answer has yet been given concerning the implication of any microorganism towards these events; hence, we hypothesize that the attribution of this phenomenon to overfishing and pollution alone is not correct. The present research attempts to give a holistic overview of the health status of the species in Greece.

## 2. Materials and Methods

### 2.1. Animal Collection

Individuals of *C. sapidus* were collected from three marine areas across the northern Greek coastlines (Figure 1). Moribund and healthy individuals were collected from the first site, i.e., Porto Lagos lagoon (Vistonikos Gulf, North Aegean), in November 2020, originating from mortality incidents, obtained from local fishermen. The second and the third samplings were performed almost at the same time in November to investigate any potential threat in the local blue crab fisheries in Thermaikos Gulf. Individuals for the second and the third samplings were obtained from two different sites within Thermaikos Gulf to perform a health assessment of the local population. In total, 48 individuals were collected from Porto Lagos lagoon (Vistonikos Gulf, North Aegean), and 40 samples each from Chalastra (Inner Thermaikos Gulf, Figure 1) and Makrygialos (Outer Thermaikos Gulf, Figure 1). An appropriate sample size was calculated using the formula $n=\frac{Z^{2}P\left(1-P\right)}{d^{2}}\;$ established by Pouhoseingholi et al. [23], where “n” is the recommended sample size, “Z” refers to the corresponding statistic level of confidence, “P” is the expected prevalence of the disease, and “d” represents the precision associated with the effect size. The water temperature during the samplings was 21 °C and 19 °C in Vistonikos Gulf and Thermaikos Gulf, respectively, while the salinity was 37 and 39 PSU, respectively. In the current research, Z was calculated as 0.95, P as 0.9, and d as 0.05, in accordance with Pouhoseingholi et al. [23].

### 2.2. Tissue Sampling

Blue crabs were anesthetized by placement on ice for a 5 min duration. Carapace width (CW) was measured and then target tissues were divided into 2 pieces for further analyses. Target tissues (heart, gills, and hepatopancreas) were collected from each crab and processed for histopathological and molecular analysis. The second half of each tissue, intended for molecular analysis, was stored at −20 °C until DNA extraction. 

### 2.3. Histopathology

Tissues intended for histopathological process were placed in Davidson solution and fixed immediately for 48 h according to Shaw and Battle [24]. After the process of dehydration through graded alcohol, the samples were embedded in paraffin wax and sectioned in a rotary microscope at 4–5 μm. Histological sections were stained with haematoxylin and eosin according to the protocol of Howard et al. [25] to monitor the health status of each individual. Finally, the histopathological study was conducted utilizing an optical microscope (iSCOPE, Euromex, Arnhem, The Netherlands) mounted with a CMEX camera (CMEX-5 Pro, Euromex).

### 2.4. Molecular Detection and Identification of Hematodinium sp. and Ciliates

Genomic DNA was extracted from a piece of tissue (25 mg) from the heart, gills, and hepatopancreas of each crab from all sampling sites. The extraction of DNA was performed using the NucleoSpin Tissue^®^ kit (Macherey-Nagel, Düren, Germany) following the manufacturer’s protocol and, specifically, the guidelines of the standard protocol for human or animal tissue and cultured cells. The purity and concentration of the extracted DNA were evaluated in a Quawell Q-5000 NanoDrop spectrophotometer (Quawell Technology, San Jose, CA, USA) and the DNA samples were stored at −20 °C until PCR application. For the identification of *Hematodinium* sp. presence, the *Hematodinium*-specific primer pair HITSF1 5′ CATTCACCGTGAACCTTAGCC 3′ and HITSR1 5′ CTAGTCATACGTTTGAAGAAAGCC 3′ [26,27] was used. This primer pair amplifies a 306 bp region of the parasite’s ITS1 region. For the ciliate detection, the primer set cil-f 5′ TGGTAGTGTATTGGACWACCA 3′ and Cil-r II 5′ TCTRATCGTCTTTGATCCCCTA 3′ developed by Lara et al. [28] was used. This primer set targets a region of approximately 645 bp of the 18S rRNA gene of the ciliate group members. PCR reactions were performed in FastGene Ultra Cycler Gradient (NIPPON Genetics EUROPE, Düren, Germany) in 20 μL final volume containing 10 μL of FastGene Taq 2 × Ready Mix (NIPPON Genetics EUROPE), 0.4 μΜ of each primer, 50 ng of extracted DNA, and PCR-grade water up to the final volume. The thermocycler program for *Hematodinium* sp. was 95 °C for 3 min followed by 37 cycles of 95 °C for 30 s, 53 °C for 30 s, and 72 °C for 30 s and a final extension step of 5 min, while for the ciliates, the annealing temperature was 50.5 °C and the extension time was 45 s. Then, 5 μL of the PCR product was subjected to electrophoresis on 1% agarose gel stained with Midori Green Advance (NIPPON Genetics EUROPE). The agarose gel was observed under UV light, and successfully amplified samples were purified using the NucleoSpin Gel and PCR Clean-up^®^ kit (Macherey Nagel, Düren, Germany) according to the kit protocol. The purified products were then sequenced in both directions using the PCR primers and the Sanger methodology. Chromatograms obtained from Sanger sequencing were visually checked and analyzed using BioEdit [29] and Finch TV 1.4.0 (Geospiza, Seattle, WA, USA). Based on BLAST comparisons and searches of the newly characterized haplotypes on the NCBI website, Maximum Likelihood phylogenetic trees were constructed for both parasites using MEGA software v.7 [30], including various haplotypes belonging to closely related taxa, obtained from the GenBank database.

## 3. Results

### 3.1. Gross Signs and Histopathological Results

Among the 128 crabs examined, *Hematodinium* sp. was detected in 112 samples, resulting in a total prevalence of 87.5%. The total prevalence was calculated based on the molecular identification of the pathogen. Specifically, 40 individuals out of 128 crabs were examined histologically and the presence of the pathogen was confirmed molecularly. For the remaining individuals investigated, molecular detection was conducted in order to calculate the total prevalence and the local prevalence. More specifically, *Hematodinium* sp. was detected with 87.5% prevalence at the Chalastra sampling site, with 89.6% and 85% prevalence in the Vistonikos Gulf and Makrygialos sampling sites, respectively. The average carapace length size was 11.8 cm, 12.5 cm, and 12.9 cm for the crabs collected from Chalastra, Thermaikos Gulf, Makrygialos, Thermaikos Gulf, and Vistonikos Gulf, respectively.

No characteristic macroscopical sign was observed during the sampling of *C. sapidus* in each case except from the characteristic “milky” coloration of the hemolymph in infested crabs. However, lethargy was observed in the specimens collected from Vistonikos Gulf in which mortality occurred. Microscopically, parasitic infestation was observed in all tissues assessed for this study (Figure 2).

Uninucleate and multinucleate parasites were observed in the gills, hepatopancreas, and heart, typical of previous histological descriptions of the *Hematodinium* sp. in crustaceans. Specifically, in the gill tissue, ameboid trophonts were detected in the gill lamellae, alongside fouling microorganisms in the space between the lamellae part of the gill (Figure 2B and Figure 3).

*Hematodinium* sp. were also detected in the heart of each infested crab in both the myocardium and pericardium parts as uninucleate trophonts and in multinucleate form with characteristic chromatin (Figure 2F and Figure 4A,B).

Hence, *Hematodinium* sp. cells have been detected in massive numbers in the hemal space of the hepatopancreas alongside several host cells (Figure 2D). Concerning the presence of the parasite in tissues and the pathology in the species of *C. sapidus*, abnormalities were mostly detected in the digestive gland. Specifically, the concentration of the hemocytes was higher in the infected specimens, which is related to the immune responses of the host to the invasion of the parasite. Additionally, the abnormal development of the epithelial cell wall was detected in infected specimens. Specimens infected with *Hematodinium* sp. demonstrated thicker epithelium in the digestive tubules, while the lumen of the digestive tubules presented structural abnormalities (Figure 5) that were not observed in healthy individuals (Figure 5A). Finally, reserve inclusion cells were scarce compared with the digestive glands of healthy individuals (Figure 5B).

### 3.2. Molecular Identification of Parasites

The molecular phylogeny of the *Hematodinium* confirmed its identity as *H. perezi*, characterizing only one identical haplotype in all analyzed specimens, which was submitted to the GenBank database and assigned the accession number PP056127. It was grouped together with several other *H. perezi* haplotypes, hosting other crab species (Figure 6). To the best of our knowledge, this is the first report of this parasite in Greece, providing evidence that it has invaded together with the blue crab.

On the other hand, the molecular analysis did not succeed in identifying the ciliate parasite at the species level, mainly because of the absence of available sequences in the GenBank database. Only one haplotype was defined in all analyzed samples, presenting more than 99.5% sequence similarity with two *Epistylis* sp. isolates (Figure 7). This haplotype was submitted to the GenBank database and given the accession number PP048746. Two other epibiotic peritrich ciliate parasite genera were very closely genetically related, namely, *Myschiston* and *Zoothamnium*.

## 4. Discussion

The Mediterranean Sea is considered to be one of the most invaded marine habitats worldwide [31]. Biological invasions tend to become more frequent each year as a result of global climate change and direct human pressures [32,33]. Global climate changes, in combination with direct impacts on marine ecosystems (temperature rise, salinity changes, ocean acidification), are key drivers in shifting the flora and fauna in ecosystems [6,34]. Hence, the impacts of global climate change tend to also affect the physiology of the hosts and the susceptibility of marine species in infectious diseases [35,36,37,38,39]. Elevated temperatures not only impact the host, but also promote the proliferation of pathogenic microorganisms and their seasonality pattern by prolonging their favored temperature [40]. Keeping in mind the downregulation of the immune responses of the host and the uncontrollable proliferation of pathogenic microorganisms, we can assume that the synergy of the aforementioned factors may result in a considerable shift in the biodiversity of marine ecosystems by favoring invasions and pressuring endemic species. Additionally, direct human pressures such as the overexploitation of marine species, commercial shipping with transfers over ballast water, and the opening of pathways to various marine ecosystems (e.g., the Suez canal) could also lead to the shaping of the diversity of species in local ecosystems [40,41,42,43].

The Atlantic blue crab, *C. sapidus*, was first recorded in Europe in 1901 on the Atlantic coast of France [11]. The invasion in the inner part of the Mediterranean Sea was documented in 1949 and was attributed to ballast waters [14]. Since then, the crab has been reported to invade marine ecosystems in the Iberian Peninsula, in the Italian Peninsula, in Greece, in Turkey, and in Morocco [11,12,37,38,39,40]. Despite the invasive nature and the general ecological traits of the species, including early maturity, rapid growth rates, opportunist diets, and a high reproduction rate, there is a lack of knowledge of this invading species regarding its interactions with the endemic benthic communities and the negative impact of its presence in local ecosystems [14,41]. Despite its territorial presence and the negative impacts that have been recorded in other species [44,45,46,47], the blue crab is considered to be an important species for fisheries in the United States of America and an emerging opportunity for fishery exploitation in Europe [14,48].

In Greece, *C. sapidus* is commercially exploited, but studies are limited in terms of ecological data and the distribution of the species in new habitats. Blue crab populations are subjected to fluctuations in numbers in almost every habitat on Greek coastlines; however, no research has been conducted to clarify the causes of these fluctuations, attributing population declines to environmental causes and the overexploitation of the species by fishermen. The current study constitutes the first report of *Hematodinium perezi*, a pathogenic dinoflagellate parasite, in Greece, as well as the ciliate *Epistylis* sp., both of which infect crustaceans and have been correlated several times with mortalities in *C. sapidus. C. sapidus* harbors a varied community of parasitic, pathogenic, and commensal microorganisms [49]. Some of these microorganisms are well studied, while others have been proposed for symbiotic traits in the host. The presence of *Epistylis* sp. is associated with environmental factors such as temperature and salinity fluctuations [49]. The proliferation of fouling microorganisms affects the physiology of the host, resulting in metabolic depression [50].

Infections with *H. perezi* are fatal and proliferate rapidly at high temperatures and salinities, and there is currently no established and effective method for managing this specific infection [51,52]. Typically, the observable indicators of infected hosts include lethargy and alterations in the hemolymph, tissues, and internal organs [43]. Advanced infection stages manifest as lethargic hosts with noticeable accumulations of parasitic stages in both their hemolymph and tissues [43]. The color of the hemolymph is frequently altered to a white/cream hue, reflecting the substantial presence of parasites [43,53]. *Hematodinium* infections may cause a profound impact on natural crustacean populations and the fisheries related to them in various ways [54,55]. Immediate consequences involve a decline in harvestable resources and a diminished recruitment rate into the fishery [43]. The parasitic dinoflagellate *H. perezi* is capable of infecting a broad spectrum of phylogenetically related crustacean hosts [46]. So far, infections attributed to *Hematodinium* or *Hematodinium*-like organisms have been documented in more than 40 species of marine crabs, shrimps, lobsters, and amphipods [43,47]. Within the coastal waters of the USA, the dinoflagellate parasite has affected a range of cohabiting wild crab species such as *Libinia dubia*, *Pagurus pollicaris*, and *Eurypanopeus depressus*, in addition to its primary host *C. sapidus* [56]. Regarding the detection of the parasitic species in China, *Hematodinium* spp. infections have been recognized as the causative agent of “milky blood disease” and “Yellow water disease” in the cultured species *Portunus trituberculatus* and *Scylla paramamosain* on Chinese coasts, respectively [57,58]. Additionally, *Hematodinium* sp. infection was identified in the shrimps *Exopalaemon carinicauda* and *Penaeus monodon* in cultured conditions in coastal areas of China [58,59,60]. *Hematodinium* sp. infections were also detected on Russian coastlines, infecting crab species such as *Paralithodes camtschaticus*, *Paralithodes platypus*, *Chinoecertes bairdi*, and *Paralithodes brevipes* [61,62]. On the Atlantic European coast, *H. perezi* infections have been reported in crab species such as *Carcinus maenas*, *Liocarcinus depurator*, *Portumnus latipes*, *Ovalipes ocellatus*, *C. sapidus*, *Cancer pagurus*, *Necorapuber*, *Pagurus bernhardus*, *Pagurus prideaux*, *Minuda rugosa*, and *Eriphia verrucose* [13,63,64,65,66,67,68]. In the inner Mediterranean Sea, *H. perezi* infections have been documented in the Italian part of the Adriatic Sea and in Akyatan Lagoon, Turkey [13,69].

The present study represents the initial documentation of *H. perezi* and *Epistilys* sp. infection in the Aegean Sea, affecting the commercially important species *C. sapidus* and leading to mortalities. The reported results from this study are in line with other studies concerning the lack of gross signs of infection even in heavily infected individuals of the crab species *C. sapidus* [70,71]. However, individuals infected with the parasitic dinoflagellate appeared more lethargic in comparison with healthy ones. Although no external sign of the disease was detected in the infected individuals, the hemolymph presented the typical “milky” coloration associated with this infection, as found in other studies [43,62]. Furthermore, the developmental stages of the parasite were recorded in almost all tissues intended for histopathological examination. More precisely, ameboid trophonts in conjunction with filamentous trophonts were observed to parasitize the myocardium and pericardium of the crab and cause the dilation of the hemal sinuses. This finding is corroborated by studies conducted on the dinoflagellate parasite in *C. sapidus* in which dilations of the heart tissue in mass parasitic infiltration were similarly observed [54,72]. Moreover, the massive infiltration of parasitic cells was documented in the hepatopancreas. A profound infestation of parasites in the hepatopancreas was found to be associated with structural alterations, including the atrophic development of the digestive tubule epithelium, which may ultimately result in epithelial lysis [62]. Structural alterations were additionally identified in the tubule lumen of the hepatopancreas in infected individuals. Specifically, the afflicted crabs exhibited dilations and irregularities in the digestive lumens of the hepatopancreas, often accompanied by the collapse of villi in numerous instances. Alterations in the lumen of the digestive tubules have been previously documented in *H. perezi* infections, revealing swelling in the tubule villi of mudflat crabs, *Helis tientsinensis* [73]. The final histopathological observation linked to the infection included a diminished count of hemocytes and reserve inclusion cells in the hepatopancreas. The current result is in line with Huang et al. [65], who observed identical results in *H. tientsinensis* infected with *H. perezi*. Regarding the pathological observation in the gills, the massive infiltration of ameboid trophonts was observed in the gill lamellas, which appeared to be swollen with deformities along the structure. The histological results are confirmed by those of Huang et al. [65], who also observed deformities in the gills associated with infections. Furthermore, the presence of parasite cells in all three tissues examined is a typical result, as *Hematodinium* sp. infection is characterized as a hemolymph disease [70,71,74].

## 5. Conclusions

Crabs are highly valuable marine resources, serving as a significant part of both commercial fisheries and marine ecosystems [75,76]. The economic importance of the crab trade is underscored by their demand in seafood markets, as is their vital role in supporting livelihoods in coastal communities by their culture or fisheries [77,78]. Nevertheless, the health status of crab populations occasionally faces threats from diverse factors, with disease emerging as a significant concern. The occurrence of mortalities due to disease carries repercussions, contributing to a decline in crab populations and resulting in diminishing productivity in fisheries. Disease outbreaks can stem from a range of pathogens, encompassing parasites, bacteria, and viruses, and can significantly affect the physiological and structural integrity of crab populations. In the current study, a pivotal discovery was made that implicated dinoflagellate *H. perezi*, for the first time, in the annual decline of the blue crab, *C. sapidus*, along the Greek coastlines of the North Aegean. While infections of *H. perezi* are common and extensively studied in numerous crab species and fishery hotspots, the fluctuations in blue crab (*C. sapidus*) populations originating from the Eastern Mediterranean coastlines remain largely unexplored. *H. perezi* is an important pathogen infecting a variety of crustacean decapods, resulting in huge mortalities [50,59]. Furthermore, the introduction of non-indigenous species acting as potential hosts for significant pathogens, such as blue crabs, may present a threat to other vital fishery species. In addition, considering the global climate change phenomenon, which can act as a driver for the further introduction of non-indigenous species, we can concur that the management plans for the mitigation of these phenomena are mandatory for the sustainability of fisheries. In recent years, blue crab mortalities in Greece, such as the mortality event that occurred in the Kotychi lagoon in Elis, remained unexplored and the phenomenon was only attributed to the increased water temperature by local authorities. In this research, the identification of an important pathogen such as *H. perezi* further highlights the fluctuations in the population of crustacean decapods in Greece.

## Figures and Tables

**Figure 1 microorganisms-12-00456-f001:**
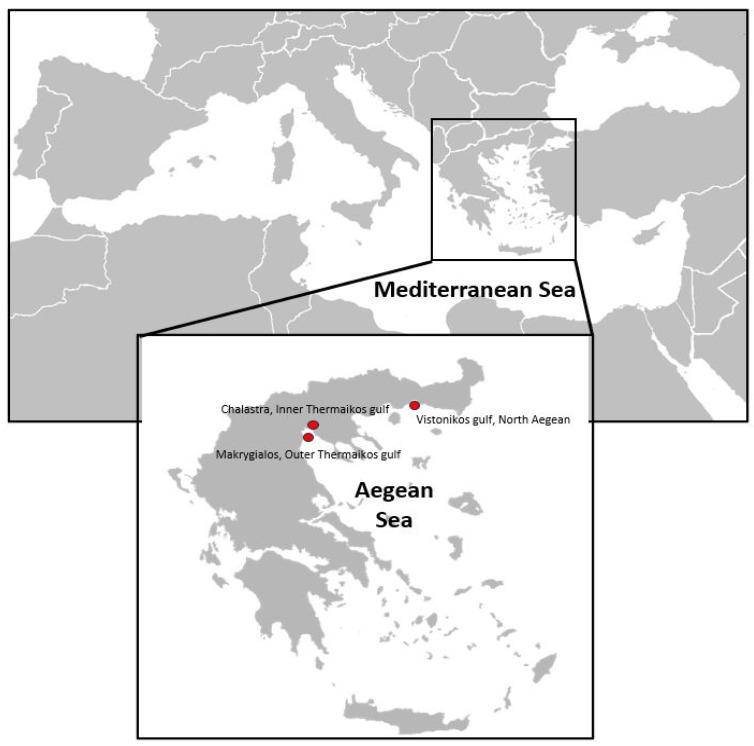
Sampling sites of *C. sapidus* individuals for the current study.

**Figure 2 microorganisms-12-00456-f002:**
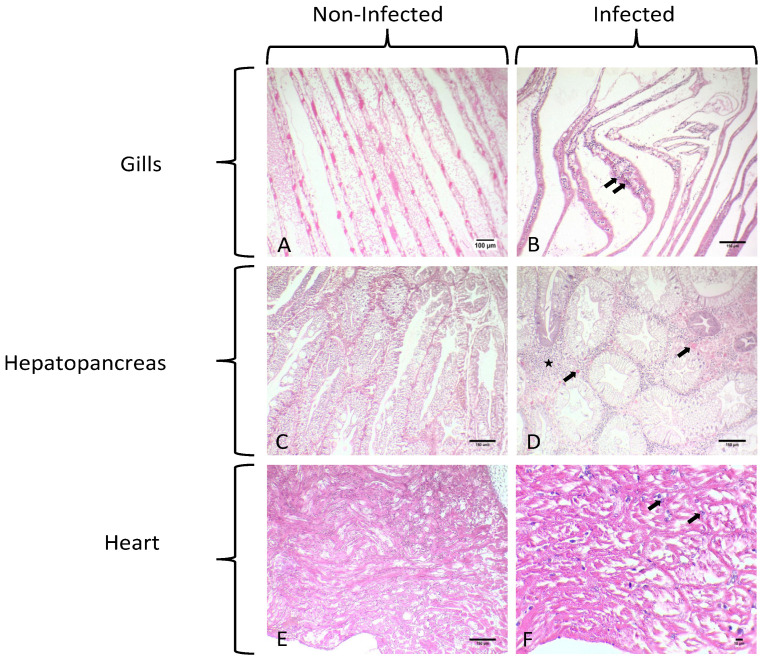
Histological sections of healthy and infected *C. sapidus* individuals collected. (**A**) Histological section of healthy gills of *C. sapidus*. (**B**) Uninucleate and multinucleate stages of *Hematodinium* sp. depicted in gills of an infected individual (arrows). (**C**) Histological display of Hepatopancreas of a healthy individual. (**D**) Histological display of hepatopancreas belonging to an infected individual. Heavy parasitism documented in hemal sinus (star) and fewer reserve inclusion cells as a result of the infestation. (**E**) Histological display of the heart of a healthy individual. (**F**) Histological micrograph of heart infected with *Hematodinium* sp. presenting dilations in the hemal sinuses. Myocardium of heart infected with *Hematodinium* cells (arrows). Total magnification in the microphotographs was ×100, ×100, ×100, ×100, ×100, and ×400 in (**A**–**E**) and (**F**), respectively.

**Figure 3 microorganisms-12-00456-f003:**
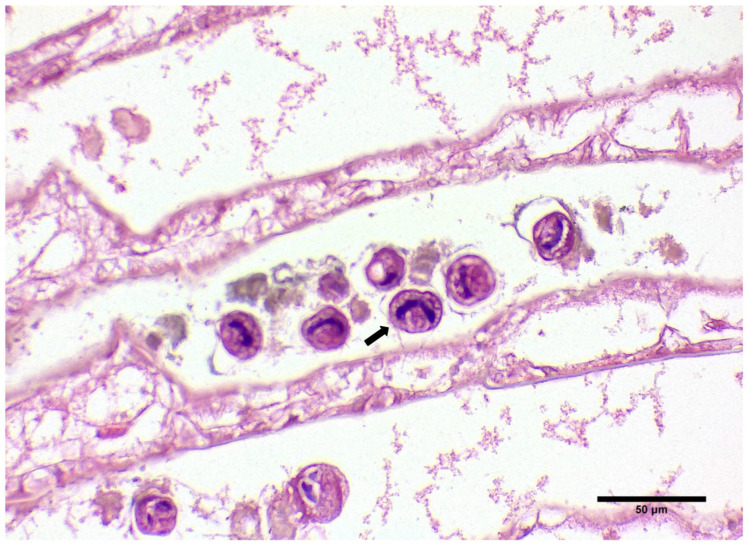
Histological display of gills infected with ciliates documented in *C. sapidus* individuals regardless of parasitism with *Hematodinium* sp. H&E staining. Total magnification ×200.

**Figure 4 microorganisms-12-00456-f004:**
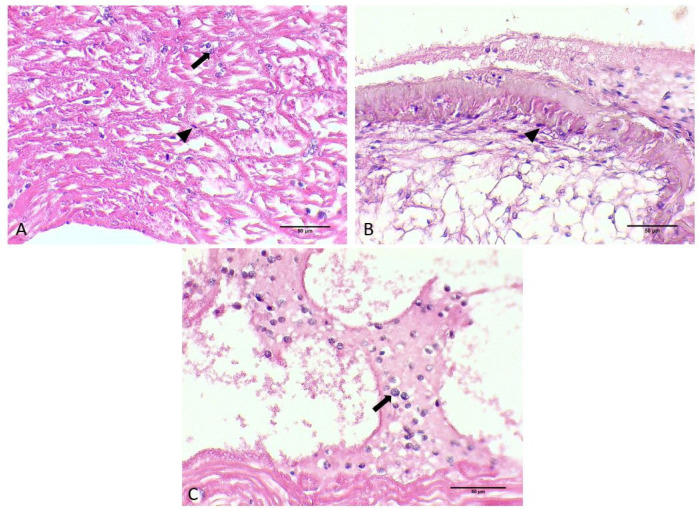
Histological sections of heart originating from *C. sapidus* individuals. Myocardium infected with *Hematodinium* sp. ameboid trophonts (arrow) and filamentous trophont (arrowhead) (**A**). Filamentous trophonts documented in pericardium of infected *C. sapidus* (**B**). *Hematodinium* sp. cells in pericardium of infected individuals (**C**). H&E staining. Total magnification ×400.

**Figure 5 microorganisms-12-00456-f005:**
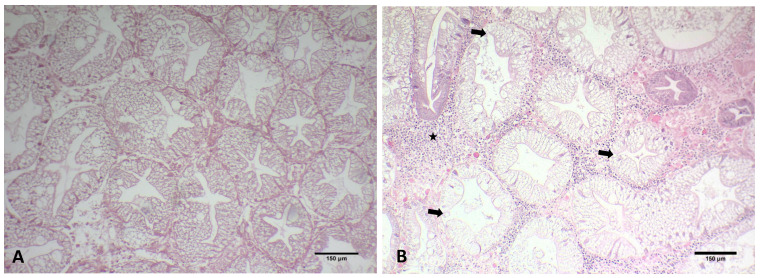
Histological display of infected hepatopancreas of *C. sapidus*. (**A**) No *Hematodinium* sp. cells were detected in the hemal space in the digestive gland and no atrophic development was observed in the digestive tubule epithelium. (**B**) Mass presence of *Hematodinium* sp. trophonts in the hemal space of infected hepatopancreas (star). Atrophic development of epithelium in digestive tubules alongside abnormalities in the structure of the lumen (arrows). H&E staining. Total magnification ×100.

**Figure 6 microorganisms-12-00456-f006:**
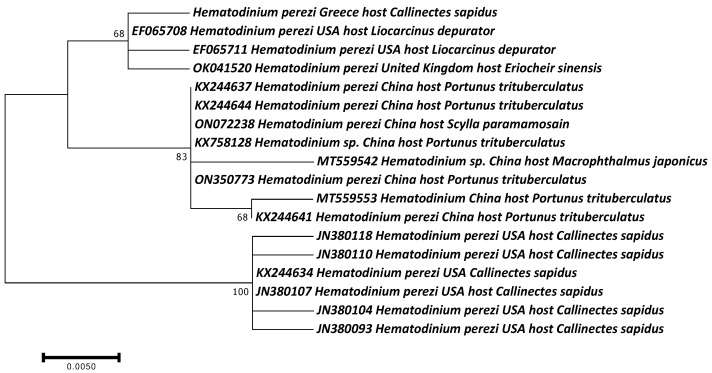
Maximum Likelihood dendrogram of the *H. perezi* haplotype found in the Aegean Sea, in comparison with other conspecific and congeneric haplotypes hosting various crab species. Bootstrap values over 60 are shown on each branch.

**Figure 7 microorganisms-12-00456-f007:**
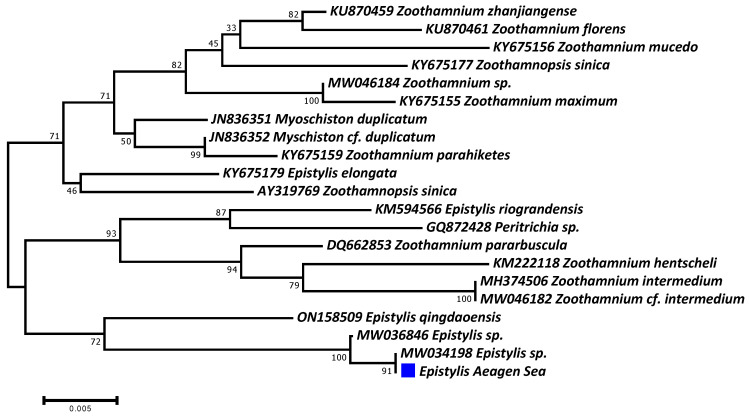
Phylogenetic dendrogram of the *Epistylis* isolate (blue square), confirming its taxonomy up to genus level.

## Data Availability

The genomic data produced are available from the GenBank database under the accession numbers PP056127 and PP048746.

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
