# Peer review of "Histopathology and Phylogeny of the Dinoflagellate *Hematodinium perezi* and the Epibiotic Peritrich Ciliate *Epistylis* sp. Infecting the Blue Crab *Callinectes sapidus* in the Eastern Mediterranean"

_microorganisms, 2024, doi:10.3390/microorganisms12030456_

Round 1

Reviewer 1 Report

Comments and Suggestions for Authors

This study found a high proportion of infected H. perezi in moribund blue crabs, which is very helpful for identifying the cause of population decline. However, the results of this paper still cannot support the conclusion. The infection of blue crabs with H. perezi and the harm caused by infection H. perezi have been reported, but it is not the latest discovery. This article only found that such infections can also occur locally. In addition, there is insufficient evidence to determine whether the main cause of a large number of dead crabs is caused by H. perezi. In addition to increasing the sample size, it is also necessary to test individuals caught in the environment, rather than only sampling moribund individuals. The author also emphasized in the article that the decline in the blue crab population is mainly caused by human activities and climate change, so there are still some issues with the writing approach of the entire manuscript. In the preface, the author extensively mentions issues such as invasion, which is not the main focus of this study and should not be overly emphasized. Suggest the author to increase the sample size and epidemiological survey data, and reorganize the writing approach.

Comments on the Quality of English Language

Pay attention to the writing style in English.

Author Response

Reviewer 1

This study found a high proportion of infected H. perezi in moribund blue crabs, which is very helpful for identifying the cause of population decline.

Response: We are grateful to the reviewer for recognition of the value of our work. We trust that after incorporating both reviewers’ recommendations, the revised manuscript has been substantially improved.

However, the results of this paper still cannot support the conclusion. The infection of blue crabs with H. perezi and the harm caused by infection H. perezi have been reported, but it is not the latest discovery.

Re: Indeed, the reviewer is correct. Infection of blue crabs with H. perezi has already been described in previous studies. Nevertheless, the scope and the novelty of our study do not stand on the description of the infection of this pathogen on the blue crab, but instead on the clarification of the blue crab population reductions and the involvement of harmful microorganisms in this phenomenon. It should be noted that, before these pathogens detections, solely overfishing and pollution were erroneously considered as responsible factors for blue crab population declines in Greek marine areas. This explanation was more clearly and in detail added in the abstract, introduction scope paragraph and conclusion (please see these sections in the revised manuscript). Furthermore, the study represents the first report of H. perezi in any marine area of Eastern Mediterranean, which is a fact of high importance as a new invasion threatening biodiversity.

This article only found that such infections can also occur locally. In addition, there is insufficient evidence to determine whether the main cause of a large number of dead crabs is caused by H. perezi. In addition to increasing the sample size, it is also necessary to test individuals caught in the environment, rather than only sampling moribund individuals.

Re: We apologize for this mistake and in the same time thank the reviewer for this point. The way these parts were written was confusing. Indeed, individuals caught in the environment had been collected as well, several of which were healthy. Both healthy and moribund animals were therefore collected from the environment by the divers. According to the reviewer’s comment, the sample size was increased by analyzing a larger number of samples that had been collected and therefore the relevant parts of Materials and Methods were accordingly modified (please see sections 2.1 and 3.1 in the revised manuscript)

The author also emphasized in the article that the decline in the blue crab population is mainly caused by human activities and climate change, so there are still some issues with the writing approach of the entire manuscript.

Re: Taking into consideration the reviewer’s comment, the abstract and mainly the scope of the study in the introduction were rewritten in an effort to clarify that the purpose was to enlighten the hypothesis that human activities (overfishing and pollution) are not the main factors for blue crab population reductions. Global climate change was only mentioned as the main driver of bio-invasions, and thus it is noteworthy and deserves pointing out in the manuscript. Based on the aforementioned, the conclusion was also accordingly rephrased.

In the preface, the author extensively mentions issues such as invasion, which is not the main focus of this study and should not be overly emphasized. Suggest the author to increase the sample size and epidemiological survey data and reorganize the writing approach.

Re: We agree with the reviewer regarding the invasions and hence a relevant sentence was added in the introduction mentioning also the threat of aquatic invasive pathogens. We believe that the revised introduction is accordingly reorganized, more focused and clear in terms of the main focus of the study. Further, as both reviewers recommended, the sample size and epidemiological survey data were increased (please see revised sections 2.1 and 3.1)

Reviewer 2 Report

Comments and Suggestions for Authors

I had the pleasure to review study titled ‘Histopathology and phylogeny of the dinoflagellate Hematodinium perezi and the epibiotic peritrich ciliate Epistylis sp. infecting the blue crab Callinectes sapidus in Eastern Mediterranean’ and based on my evaluation, it is an important study and well written manuscript. However, authors might consider addressing the following concerns: 

1/ On the tissue sampling section for how long was the blue crabs anesthetized by placement on ice? This should be explained. 

2/ There is use of prevalence and prevalence rate interchangeably and needs to be consistent. 

3/ There is no sample size calculation method in the study, and it should be incidence rate rather than prevalence or prevalence rate. These epidemiological terms should be explicitly used in the paper. 

4/ Which type of microscope was used for the histopathology lesions and what magnification? 

5/ Was the prevalence calculated based on the HE staining or molecular tool? Please specify. 

6/ From Figure 5, how can we proof it’s an infection site from the image? Do you have control for this? Or based on what peculiar features was the decision made? 

7/ Any PCR performed to detect the species? 

8/ Was any institutional ethical approval given to perform this study ? 

Author Response

Reviewer 2

I had the pleasure to review study titled ‘Histopathology and phylogeny of the dinoflagellate Hematodinium perezi and the epibiotic peritrich ciliate Epistylis sp. infecting the blue crab Callinectes sapidus in Eastern Mediterranean’ and based on my evaluation, it is an important study and well written manuscript. However, authors might consider addressing the following concerns:

Response: We feel grateful to the reviewer for the very kind words regarding our study. We believe that the revised version has addressed all the suggested concerns

1/ On the tissue sampling section for how long was the blue crabs anesthetized by placement on ice? This should be explained.

Re: Blue crabs were anesthetized for five minutes. This info was added in section 2.2 in the revised manuscript, in accordance to the reviewer’s comment.

2/ There is use of prevalence and prevalence rate interchangeably and needs to be consistent.

Re: Prevalence was only kept in the whole revised manuscript for consistency, as recommended by the reviewer.

3/ There is no sample size calculation method in the study, and it should be incidence rate rather than prevalence or prevalence rate. These epidemiological terms should be explicitly used in the paper.

Re: Based on a relative comment of the first reviewer, more samples that had been already collected from the same marine areas were analyzed and the formula documented by Pouhoseingholi et al. 2013 was applied for the proper epidemiological sample size calculation, as recommended by the reviewer (please see sections 2.1 and 3.1 in the revised manuscript). Taking into consideration the increase of the sample size, the term “prevalence” is now appropriate rather than incidence.

4/ Which type of microscope was used for the histopathology lesions and what magnification?

Re: Following the reviewer’s comment, details for the type of microscope and total magnification were added in section 2.3 of the revised manuscript, as well as in the legends of Figures 2-5.

5/ Was the prevalence calculated based on the HE staining or molecular tool? Please specify.

Re: Prevalence was calculated based on the molecular results. This was added in detail in section 3.1 for clarification, in accordance to the reviewer’s comment.

6/ From Figure 5, how can we proof it’s an infection site from the image? Do you have control for this? Or based on what peculiar features was the decision made?

Re: We agree with the reviewer’s point. Infection sites were compared with healthy individuals, as added in the end of 3.1 section, whereas Figure 5 was replaced by a new one with comparison to a healthy individual’s image (please see Figures 5A and 5B and their legend in the revised manuscript).

7/ Any PCR performed to detect the species?

Re: We are not sure we understand correctly the comment of the reviewer. Sections 2.4 and 3.2 describe in detail the PCRs performed for the molecular identification of the two pathogens. On the one hand Hematodinium perezi was successfully and reliably identified at species level, but on the other hand Epistlylis was only identified up to genus level on account of absence of characterized species in the GenBank database. Both newly described sequences were deposited in the GenBank database (please see details in section 3.2)

8/ Was any institutional ethical approval given to perform this study?

Re: We thank the reviewer for this final comment. It should be noted that all specimens were purchased from professional anglers. Also since blue crabs belong to crustaceans, no further ethical approval was needed for this study.

Round 2

Reviewer 1 Report

Comments and Suggestions for Authors

no

Author Response

We would like to thank the reviewer for the positive comments.